# Effect of Heat Treatment on Microstructure and Mechanical Properties of the AZ31/WE43 Bimetal Composites

**Dexing Xu [1], Kangning Zhao [1], Changlin Yang [2] 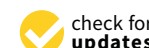, Hongxiang Li [1],\* and Jishan Zhang [1],\***

[1]   State Key Laboratory for Advanced Metals and Materials, University of Science and Technology Beijing, 100083 Beijing, China; xudexingstar@163.com (D.X.); b20140492@xs.ustb.edu.cn (K.Z.)

[2]   State Key Laboratory of Solidification Processing, Northwestern Polytechnical University, Xi'an 710072, China; ycl@nwpu.edu.cn

\*   Correspondence: hxli@skl.ustb.edu.cn (H.L.); zhangjs@skl.ustb.edu.cn (J.Z.); Tel.: +86-10-62332350 (H.L.)

**Abstract:** Effect of heat treatment on the microstructure and mechanical properties of the AZ31/WE43 bimetal composites was investigated, and the optimized solution treatment and ageing treatment parameters were achieved. After heat treatment, it was found that that the high melting-point Y-rich phases and Mg-RE (rare earth) phases at the interface were distributed more uniformly, so the interfacial strength and plasticity were improved. Meanwhile, the shear strength and plasticity of AZ31 were increased, and the shear strength of WE43 was also improved by heat treatment. The evolution of interface morphologies and enhancement of the mechanical properties at the interface will be discussed in detail.

**Keywords:** Mg/Mg bimetal composites; heat treatment; interfacial microstructure; mechanical properties; fracture behaviors

## 1. Introduction

Magnesium alloys are becoming increasingly attractive for potential applications in aerospace, automobile, and electronics, due to the advantages of light weight, high specific strength, and high specific stiffness [1–3]. However, the conventional magnesium alloys, such as AZ31, AZ61, and ZK60, have many undesirable properties, such as poor wear resistance and relatively low mechanical strength, which has restricted their wide application in many industrial fields [4,5]. On the other hand, Mg-based alloys containing rare earth (RE) elements, such as WE43, WE54, and GW103, are achieving ultra-high strength, due to the combination of solid solution strengthening and precipitation hardening [6–8]. However, the high cost and poor toughness of Mg-RE alloys greatly limit their range of applications. In order to save the manufacturing cost and improve the material properties, a new structural design for creating laminated composites with excellent ductility and high strength, and decreasing the amount of RE elements, has been intended. Thus, an Mg/Mg hybrid metal is a desirable lightweight composite that benefits from the advantages of both heavy rare earth Mg alloys and commercial high-strength Mg alloys. The Mg/Mg bimetallic materials can be achieved by ECAE (equal channel angular extrusion) [9], ARB (accumulative roll bonding) [10], compound casting [11,12]. Recently, a new AZ31/WE43 bimetal composite with a good interfacial metallurgical bonding was fabricated using an insert molding method by the authors [11]. Moreover, the as-cast microstructure and bonding mechanism at the interface have also been investigated. However, although good metallurgical bonding was achieved for the Mg/Mg bimetal composites, they have poor overall properties because of uneven microstructures and casting defects in the interface and/or basal metals.

In contrast to the as-cast studies, recently investigating the interfacial microstructure and mechanical properties of the bimetal composites after heat treatment is becoming a focus, because it can improve the interfacial bonding strength and the overall properties of bimetal composites. Dezellus et al. [13] investigated the effect of T6 heat treatment on the mechanical properties of Ti/Al–7Si–0.3Mg bimetals, and found that the size and number of Si particles decreased significantly at the interface, and the interfacial strength was improved. Liu et al. [14] reported the microstructure evolution and mechanical properties of 6101/A356 bimetal composites after T6 heat treatment, and identified that the morphology of silicon particles was changed from long, coarse plate-like, to fine spherical shape, and Zn, Mg elements were diffused more homogeneously across the interface. Thus, the interfacial shear strength can be improved by about 30%. Yan et al. [15] investigated the microstructure evolution and mechanical properties of 7050/6009 bimetal slabs during homogenization annealing, and developed an optimal homogenizing annealing process for subsequent processing. Wang et al. [16] characterized the microstructure evolution at the bonding interface of Al/Al–Mg–Si alloy after the heat treatment, similarly leading to the improved interface bonding strength. Li et al. [17] investigated the diffusion behavior of the alloying elements of 2024/3003 gradient composite at 768 K for 45 min, and then aged at 458 K for 12 h, and discovered that it included two kinds of diffusion behaviors, i.e., long-distance diffusion and short-distance diffusion at the interface with a heat treatment.

From the above studies, it is seen that heat treatment is an effective method to improve the overall properties and interfacial strength of bimetal composites. However, the heat treatment process of the Mg/Mg bimetal composites is difficult to control, even if it is so significant. During the heat treatment process, the effect of the elemental diffusion, the solution, and precipitation of interface phases on the interfacial properties, is rather important [13,18]. Moreover, it is necessary for optimizing parameters to achieve a desirable overall property, by considering the consistency between basal metals and interface layers [18]. Thus, as for the microstructure evolutions and mechanical properties at the interface of Mg/Mg bimetal composites after heat treatment, no studies have been reported thus far. For this purpose, the AZ31/WE43 bimetal composites are selected as an example, and the heat treatment process of this kind of bimetal composite is optimized. Meanwhile, the microstructure and mechanical properties after heat treatment were analyzed in detail. According to studies, the microstructure and mechanical properties of WE43 would be strengthened during heat treatment, and T6 treatment is an effective way [6,19]. However, AZ31 has no obvious solid solution and ageing strengthening effect, due to the low content of alloying elements [20]. Therefore, the heat treatment process of WE43 is used as a reference for optimizing that of the AZ31/WE43 bimetal composite, and the solution time is fixed at 12 h, in order to shorten the process time. Eventually, an optimized heat treatment parameter is selected to improve the interfacial uniformity and the overall properties of this novel bimetal composite. The study can offer useful instruction for the heat treatment process of Mg/Mg bimetal composites, with good industrial application prospects.

## 2. Experimental Procedures

### 2.1. Specimen Treatment

The principle of Mg/Mg bimetal composite material produced by insert molding was introduced in our previous works [11] in detail. Commercially available AZ31(Mg–3Al–Zn) magnesium alloy and WE43 (Mg–4Y–3RE–0.5Zr) alloy were used as casting material and solid insert material. Before the inset molding procedure, the inserts were cut into cylindrical bars with 40 mm diameter and 90 mm height. The surface of the WE43 bars was mechanically polished with SiC papers before use and then treated by alkaline cleaning, acid pickling, and ultrasonically degreasing with acetone. Similarly, the AZ31 ingots were cut into pieces and pre-treated by the same procedure, except mechanical polishing. Afterwards, the pre-treated AZ31 pieces were melted in the crucible located in a resistance furnace (SG-10-10, Jinan Changda Thermal Industry Co., Ltd., Jinan, China). During the melting process, a protective gas mixture containing $SF_6$ and $CO_2$ were used for preventing oxidation. When the melt reached

the preset temperature of 933 K, WE43 bars were immersed into the AZ31 melt, and kept for 110 s. After that, the entire assembly containing mold was taken out of the furnace and cooled to room temperature. A detailed description of the insert molding procedure and the experimental device is given in reference [11]. In this study, T6 treatment refers to a solution treatment at 673–793 K for several hours, followed by quenching and artificial ageing treatment. The samples were firstly treated under an argon atmosphere and the procedure of solution treatment was designed as 698, 723, 748, 773 and 788 K, with a fixed time of 12 h, followed by water-immersion quenching with a maximum delay of 5 s. Shortly thereafter, the specimens were subjected to an artificial ageing treatment at 483 K for different times, and quenched by cooling water.

### 2.2. Materials Characterization

To analyze the interfacial formation, the specimens were cut from the middle part of the samples parallel to the cylindrical insert with a size of 15 mm × 15 mm by setting the interface in the middle. The microstructures of the specimens were observed using a ZEISS EVO-18 (Jena, Germany) scanning electron microscope (SEM) equipped with an energy-dispersive X-ray spectrometer (EDS, Thermo scientific, Madison, WI, USA). The elemental distributions were analyzed by JEOL JXA-8230 (JEOL, Tokyo, Japan) electron probe microanalyzer (EPMA). The precipitated phases on bonding interface were also characterized by Rigaku D/max 2400 X-ray diffractometer with Cu Kα target and a Tecnai G2 F30 (FEI, Boston, MA, USA) transmission electron microscope (TEM) operated at 300 kV, respectively. The interfacial shear test was performed using a self-defined method in a universal testing machine, which was described in reference [11]. Interfacial shear strength ($\tau$) can be calculated using Equation (1) introduced in reference [11]. The Vickers hardness was measured across the interface by the microhardness tester at a load of 50 g and a dwell time of 15 s.

### 3. Results

### 3.1. Solution-Treated Microstructure

As expected, the elemental mixing and diffusion at the interface of AZ31/WE43 bimetal composites can accelerate the formation of sound metallurgical bonding. There is a continuous transition region between the AZ31 matrix and the WE43 insert. The interface is taintless and free from oxide inclusions, pores, and other undesirable defects. As shown in Figure 1a, a relatively uniform transition interface layer with a thickness of 150–180 μm consists of α-Mg grains and other intermetallic phases formed due to the contact between the WE43 alloy insert and AZ31 melt.

Figure 1b–f shows the interfacial microstructure evolution of AZ31/WE43 bimetal composite at different solution temperatures of 698 K, 723 K, 748 K, 773 K, and 788 K for 12 h. It can be seen from Figure 1b–f that the interfacial thickness after solution treatment increases from 316 μm to 782 μm, because of the long-distance diffusion of Mg, RE, Al, Zn element with increasing annealing temperature, from 698 K to 788 K. It is found that RE-rich eutectic phases at WE43 side were dissolved gradually, and finally disappeared with the increase of solution-treatment temperature. When the temperature reached 773 K, almost all of the RE-rich eutectic phase and $Mg_{17}Al_{12}$ phase were dissolved into the α-Mg matrix, and only the high melting-point white particles were evenly distributed in the interface layer. Figures 2–4 present the concentration mappings of Mg, RE (Y, Gd, Nd), Al, Zn, and Mn across the interface under the temperatures of 723 K, 773 K, and 788 K for 12 h. When the solution temperature was 723 K, the amount of eutectic phase at the interface was reduced and partially dissolved, as shown in Figure 2. When the solution temperature was above 773 K, eutectic compounds in the interface were almost dissolved, as shown in Figures 3 and 4, and the elements were also evenly distributed. When solution heat treatment is carried out at 773 K or 788 K for 12 h, the coarse RE-rich phases in the interface adjacent to WE43 undergo fragmentation, necking, spheroidization, and solution. The lamellar-like (Al, Mg)$_2$RE phase in the interface adjacent to the AZ31 side becomes

more uniform and finer. Meanwhile, the transition region of AZ31/WE43 bimetal composite become smooth and no evident gradient exists.

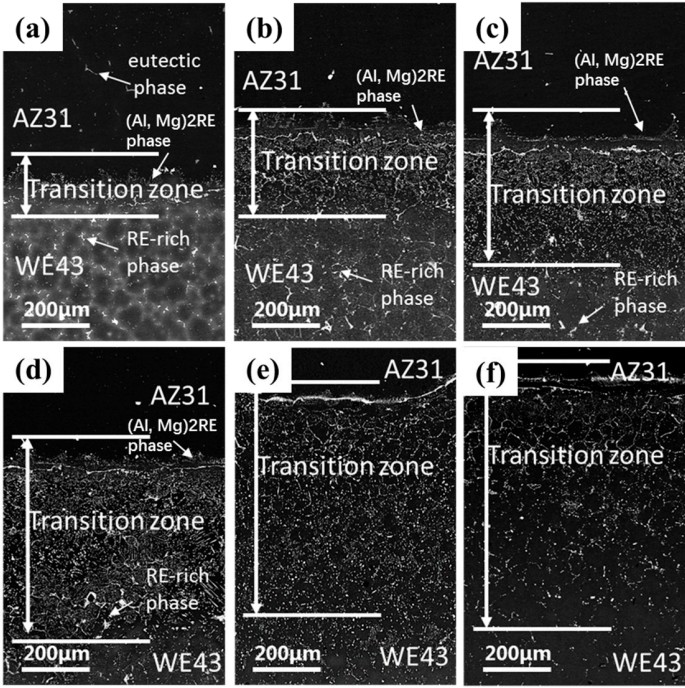

**Figure 1.** Microstructure in the interfacial region of AZ31/WE43 bimetal after different solution treatment and corresponding thickness of the transition zone: (**a**) As-cast, 185 μm; (**b**) 698 K, 12 h, 316 μm; (**c**) 723 K, 12 h, 443 μm; (**d**) 748 K, 12 h, 575 μm; (**e**) 773 K, 12 h, 681 μm, (**f**) 788 K, 12 h, 782 μm.

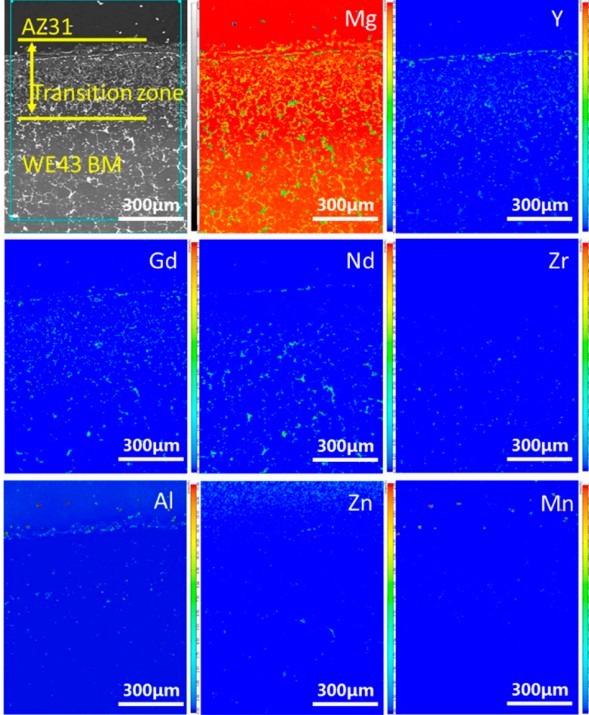

**Figure 2.** Concentration mappings of Mg, RE (Y, Gd, Nd), Al, Zn, and Mn under the solution treatment conditions of 723 K for 12 h.

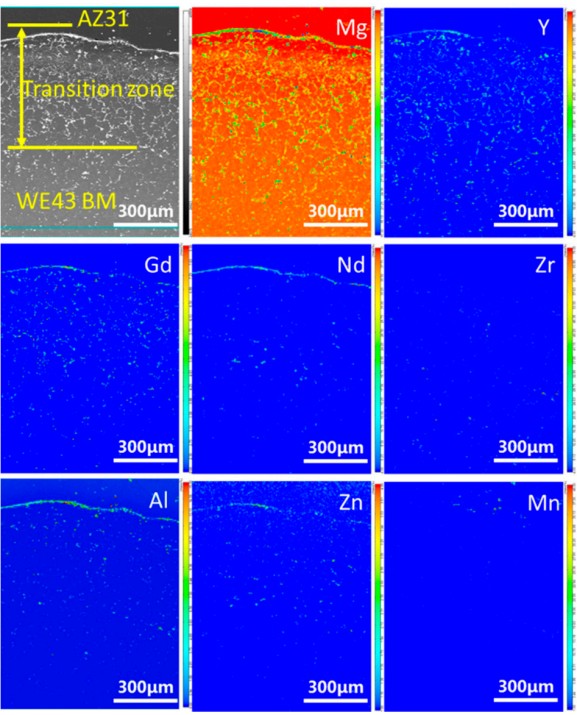

**Figure 3.** Concentration mappings of Mg, RE (Y, Gd, Nd), Al, Zn, and Mn under the solution treatment conditions of 773 K for 12 h.

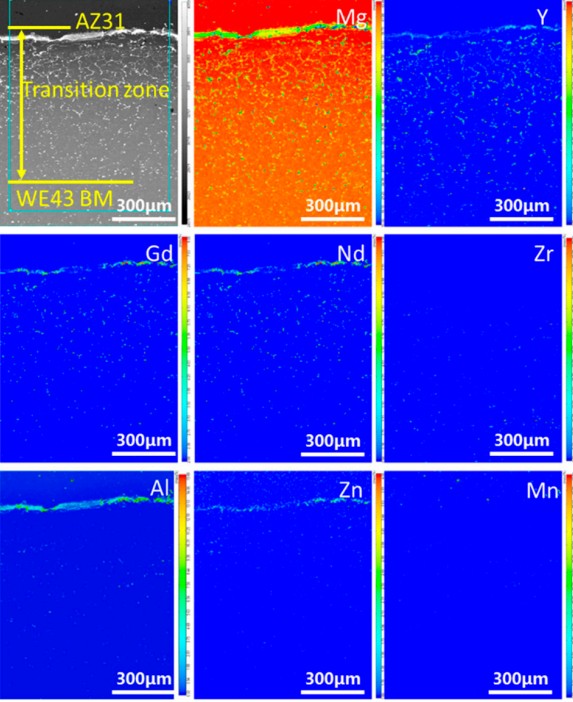

**Figure 4.** Concentration mappings of Mg, RE (Y, Gd, Nd), Al, Zn, and Mn under the solution treatment conditions of 788 K for 12 h.

It is known that the other factor that determines the solution treatment process of the bimetal composite is the dissolving degree of phases in AZ31 and WE43 matrix. It can be seen from Figure 1b–f that the $Mg_{17}Al_{12}$ phases in the AZ31 side are eliminated when the temperature was above 698 K, and no significant changes were found for the microstructure of AZ31 side, even when the temperature

was increased to 788 K, particularly without overheating. Meanwhile, the microstructure at WE43 side shows no significant evolution until the temperature reaches 773 K. The RE-rich eutectic phase almost disappears under the solution conditions of 788 K for 12 h. From Figure 4, eutectic compounds were all dissolved, no matter the interface or matrix and only some high melting-point phases are evenly distributed in the interface and WE43 matrix. By XRD analysis, the high melting-point phases at the interface after solution treatment are Y-rich phases and other Mg-RE cuboid-shaped phases, as shown in Figure 5. The high melting-point phases at the interface are further analyzed by TEM. The TEM images with the corresponding selected area electron diffraction (SAED) patterns are shown in Figure 6. From Figure 6a,c, it is found that the distribution of high melting-point phases mainly includes two different types, i.e., one is uniformly dispersed in the grain interior, and another one is clustered on the grain boundary. From Figure 6b,d, these cuboid-shaped phases are identified to be a face-centered cubic (fcc) structure, and their chemical composition is identified as the Y-rich phase. From the above results, the optimized solution treatment process of the AZ31/WE43 bimetal material can be determined to be 788 K for 12 h, at which the bimetal composite shows more uniform microstructure, no matter at the interface or in the WE43 matrix.

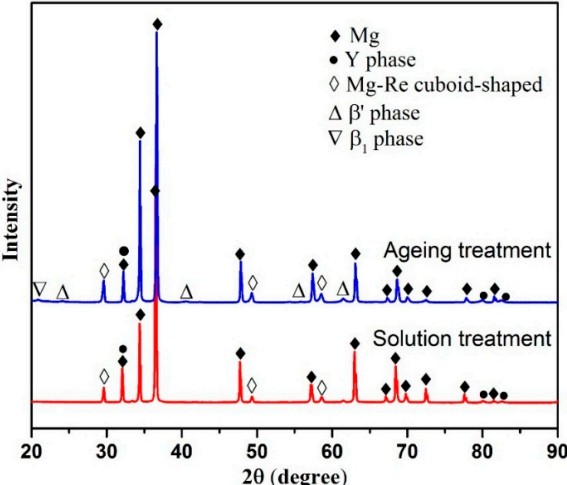

**Figure 5.** XRD patterns of solution-treated (788 K for 12 h) and ageing-treated (483 K for 48 h) specimens at the transition interface.

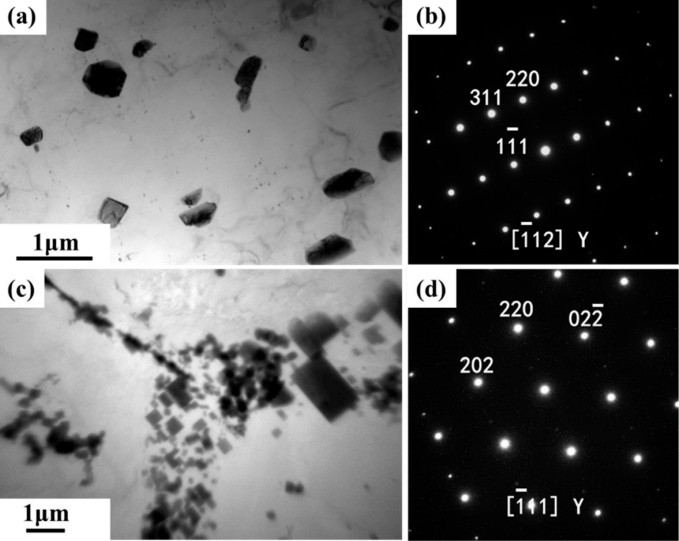

**Figure 6.** (**a**,**c**), TEM bright field images of the high melting-point phases at the transition interface; (**b**,**d**), the corresponding SAED patterns.

### 3.2. Ageing Hardening Response and Microstructure Evolution

Figure 7 shows the age hardening curves of the interface zone, AZ31, and WE43 matrix after solution treatment at 788 K for 12 h, and aged at 483 K for different hours. The shape of the ageing curves shows different characterizations for the matrix materials and interface layer of the bimetal composite. The hardness of AZ31 is almost constant in the whole ageing range, which cannot be strengthened. The hardness of WE43 increases before 36 h, and then shows a wide peak-aged platform after 36 h, indicating an excellent thermal stability. Compared with WE43, the hardening rate of interface zone is faster, and the peak value can be reached at 24 h and, then, after peak-ageing, the value decreases again. The maximum hardening value at peak-ageing is about HV100. To analyze precipitates after ageing treatment, XRD and TEM analysis of the samples at the interface were checked. From Figure 5, some small peaks marked "$\triangle$" and "$\bigtriangledown$" appear in the XRD pattern, which suggests that some new precipitates are formed in the aged specimens. Figure 8 shows the TEM micrographs with the corresponding SAED pattern, and the HTEM micrographs of precipitates after ageing at 483 K for 48 h. As seen in Figure 8a, the lens-shaped precipitates with a diameter about 20–80 nm are dispersed in the interface. From the SAED taken from the $[2\,\overline{1}10]$ zone axis, and the HTEM image shown in Figure 8c, it can be confirmed that the precipitates are β′ phases and α matrix, i.e., $(100)_{\beta'}//(1\overline{2}10)_{\alpha}$, $[001]_{\beta'}//[0001]_{\alpha}$. It was identified as an orthorhombic structure and a $= 2a_{Mg} = 0.640$ nm, b $= 8b_{\{10\,\overline{1}0\}Mg} = 2.223$ nm, c $= c_{Mg} = 0.525$ nm [19,21]. Figure 8b shows the plate shaped precipitates with a length of 100–500 nm, and the corresponding SAED pattern with the [0001] zone axis, indicating that the precipitates are $\beta_1$ phases, with a fcc structure (a $= 0.74$ nm), and the orientation relationship between $\beta_1$ and α matrix of $(\overline{1}12)_{\beta_1}//(10\overline{1}0)_{\alpha}$ and $[110]_{\beta_1}//[0001]_{\alpha}$ [19,21].

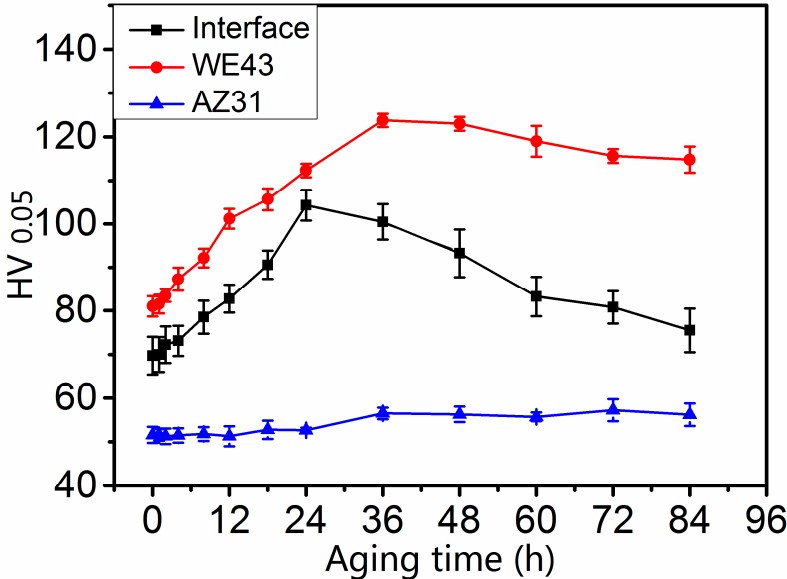

**Figure 7.** The ageing hardening curves of WE43, AZ31, and the interface.

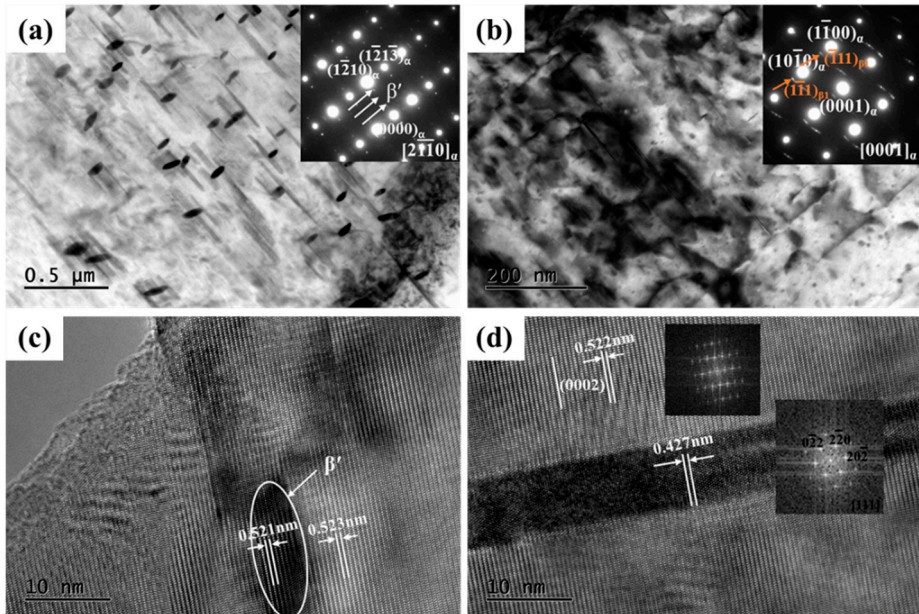

**Figure 8.** TEM analysis of the precipitates at the interface after ageing treatment (483 K for 48 h) (**a**) TEM image of the lens-shaped precipitates with corresponding SAED patterns; (**b**) TEM image of the plate-shaped precipitates with corresponding SAED patterns; (**c,d**) corresponding HTEM image.

*3.3. Mechanical Properties after Ageing*

The effect of artificial ageing time on mechanical properties was investigated with reference to the hardness analysis. As shown in Figure 9a, the shear mechanical properties show a good consistency with hardness variation as the ageing time prolongs. The shear strength value of the WE43 increases rapidly before 24 h and, then, slowly increases after 24 h. When the ageing time reaches 48 h, it has a maximum shear strength value of 221 MPa for the WE43 matrix. The shear strength value of the interface reaches a maximum value of 132 MPa at the peak-ageing condition, and there is no significant change for the shear strength after ageing treatment. When the ageing time increases to 48 h, the shear strength value of the interface decreases a little to about 120 MPa. However, the shear mechanical properties of AZ31 matrix is substantially unchanged, which is consistent with the trend of hardness variation.

Aiming at the comparison of the mechanical behaviors of AZ31/WE43 bimetal composites for different heat treatment states, three typical displacement-loading force curves, which covered solution state (0 h), peak-aged state of interface zone (24 h), peak-aged state of WE43 value, but over-ageing state of interface zone (48 h), were selected. As shown in Figure 9b–d, the displacements of the interface layers and WE43 matrix decrease with the increase of ageing time, while the AZ31 matrix is reversed. The change of displacement also reflects the evolution of plasticity to a certain extent. The plasticity of WE43 decreases as the time prolongs but the shear strength increases. The shear strength of interface zone is slightly improved without the dramatical decrease of plasticity. The shear strength of AZ31 matrix remains a constant while the plasticity is enhanced. By the above analysis, ageing treatment can improve the mechanical properties of AZ31/WE43 bimetal composites. In order to achieve the overall properties of the AZ31/WE43 bimetal composite, i.e., combining the high strength of WE43 and the high plasticity of AZ31, the optimized artificial ageing parameter can be defined to be 483 K for 48 h.

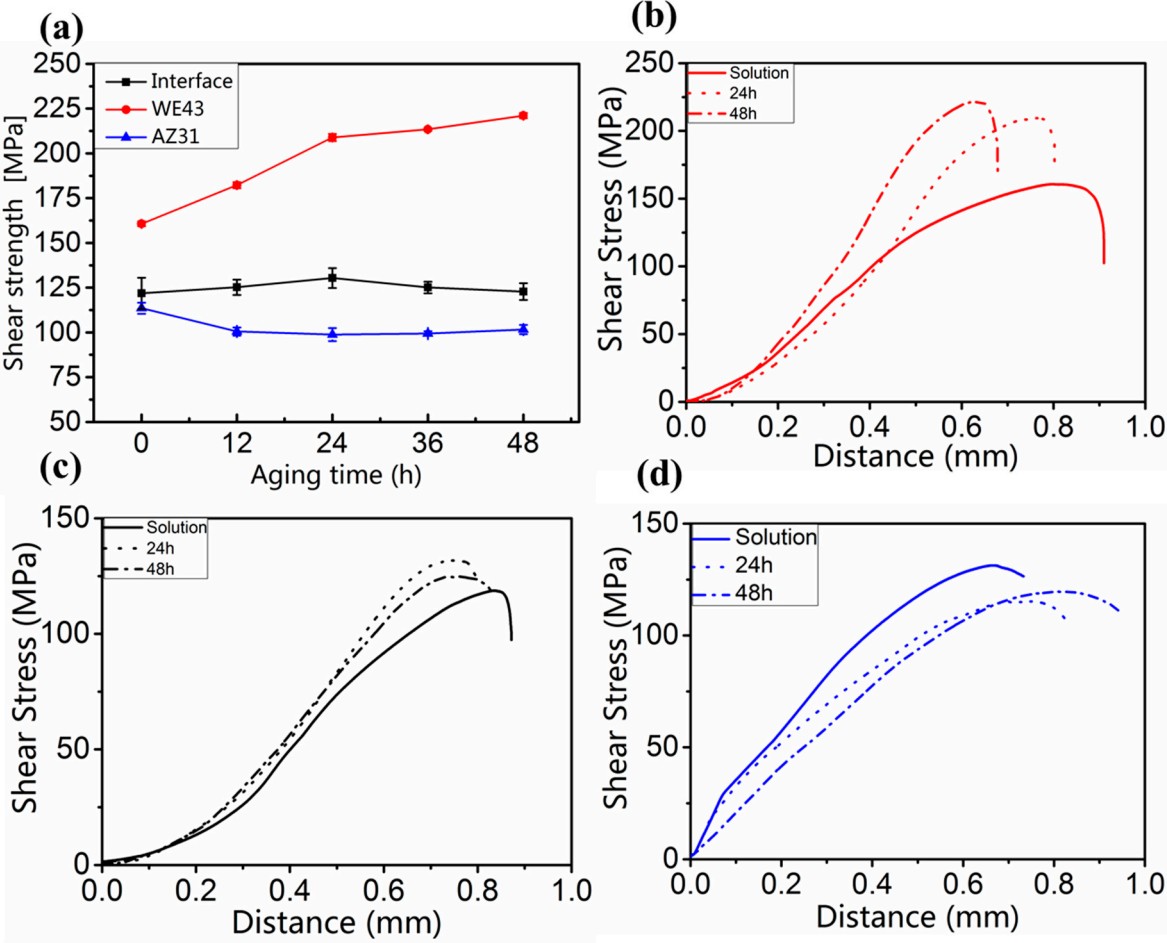

**Figure 9.** (**a**) The shear properties of the interface, WE43 and AZ31; (**b**–**d**) three typical displacement-loading force curves covering solution state (0 h), peak-aged state of interface zone (24 h), and peak-aged state of WE43 value, but over-ageing state of interface zone (48 h): (**b**) the typical displacement-loading force curve of WE43; (**c**) the typical displacement-loading force curve of interface zone; (**d**) the typical displacement-loading force curve of AZ31.

### 3.4. Fracture Behaviors at the Interface Layer

Figure 10 shows the SEM micrographs of interfacial fracture surfaces for the solution-treated samples. From Figure 10a,b, it shows that the cracks are generated in the zone with a stress concentration parallel to the load direction, and propagated with the increase of shear strength. From Figure 10c, the shear fracture surface is composed of cleavage planes. Moreover, there are some white phases distributed surrounding the cleavage planes. From Figure 10d, many small dimples can be observed near the white phases.

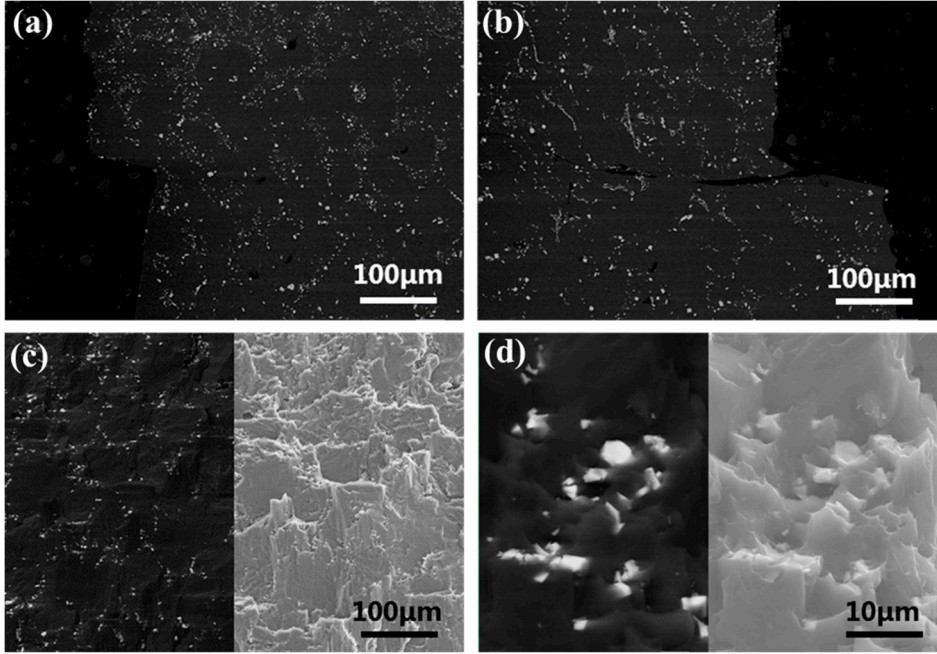

**Figure 10.** Failure sequence occurring on the solution-treatment AZ31/WE43 interface of the shear specimen. (**a**) A side face view of the failed interface between AZ31 and WE43 fracture surface indicating crack arrest lines and the initiation of cracking along the lamellar-like zone. (**b**) High magnification views of the location of cracks. (**c**) The SE and BSE images of the macroscopic appearance of the fracture surface and (**d**) high magnification views.

Figure 11 shows the SEM micrographs of interfacial fracture surfaces for the aged-treated samples. Figure 11a,b represent the cross-section image after shear fracture of T6-treated AZ31/WE43 bimetal composites. A side face view of the failed interface between AZ31 and WE43 fracture surface indicates that a main crack can initialize in the WE43 matrix, and then propagate along the interface zone. Figure 11c–f show the secondary electron (SE) and back scattering electron (BSE) images for the interfacial fracture. Figure 11c indicates that the fracture mode is mainly cleavage fracture with river patterns and white phases that are evenly distributed surrounding the cleavage planes. Figure 11d, shows another type of fracture mode, i.e., quasi-cleavage fracture with tear ridge. The tear ridge along grain boundary included some high melting-point phases, shown in the BSE images. The magnified view of the grain interior is shown in Figure 11e, in which some small cleavage planes and steps are observed. Similarly, from the magnified view of the tripe junctions at grain boundary shown in Figure 11f, some kink belts and dimples are detected. That is to say, the interfacial fracture mode of T6-treated AZ31/WE43 bimetal composites is the mixed mode of the cleavage and quasi-cleavage fractures.

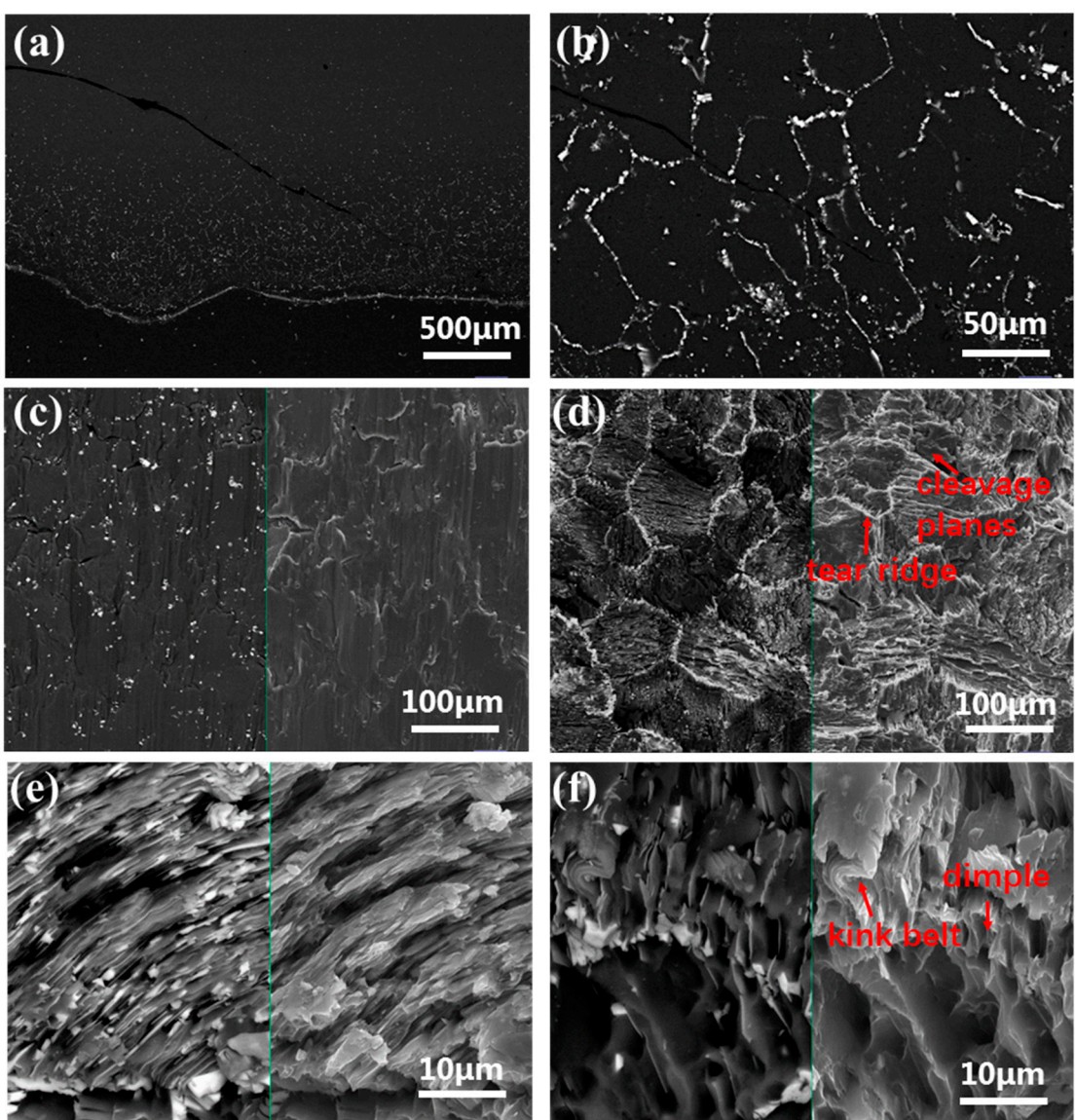

**Figure 11.** Failure sequence occurring on theT6-treated AZ31/WE43 interface of the shear specimen. (**a**) A side face view of the failed interface between AZ31 and WE43 fracture surface indicating crack arrest lines and the initiation of cracking. (**b**) High magnification views of the location of cracks. (**c,d**) The SE and BSE images of the fracture surface macroscopic appearance. (**e,f**) High magnification views of the Figure 10d.

## 4. Discussion

In our previous study [11], the solidification of the interface zones and bonding mechanism have been clarified. Due to the elemental mixing in the fused WE43 surface layer and the elemental diffusion during the solidification, the interface zone tends to form Mg–Al-RE systems. The study reported that the transition interface is mainly composed of lamellar-like and particulate $Al_2RE$ phases, $Mg_{24}Y_5$ eutectic phases, coarse Y-rich phases, and $Mg_{17}Al_{12}$ eutectic phases. A good transition interface is formed, and the average interface shear strength is 108 MPa. However, the distribution of these phases at the transition interface is not uniform, and the properties are not stable. Thus, it is necessary to optimize the interfacial microstructure and properties by heat treatment.

### 4.1. Strengthening Mechanism after Heat Treatment

During the solution treatment, dissolution of intermetallic compounds, homogenization of dendrite, and diffusion of alloying elements will occur. The optimal solution parameter of 788 K for 12 h is chosen to gain a more uniform microstructure for the bimetal composites. After solution treatment, the long-distance diffusion of RE (Y, Gd, Nd), Al and Zn across the interface would broaden the transition interfacial zone. The $Mg_{17}Al_{12}$ and $Mg_{24}Y_5$ eutectic phases at the interface are dissolved due to the low eutectic point of 709 K and 839 K, while coarse RE-rich phases at the interface undergo fragmentation, necking, spheroidization, and solution, and then distribute uniformly at the interface. Moreover, some new cuboid-shaped Mg-RE phases are generated during solution treatment. By solution treatment, the plasticity is increased due to the dissolution of the eutectic phases, and the homogenous distribution of the high melting-point Y-rich phases and Mg-RE phases, and shear strength is also enhanced, due to the solution strengthening of Al and Y. Meanwhile, it is noted that the shear strength of AZ31 is improved about 40 MPa after solution treatment at 788 K for 12 h, due to the disappearance of the eutectic $Mg_{17}Al_{12}$ phase and the reduction of casting defects.

During the ageing treatment at 483 K for different times, these high melting-point RE-rich phases and Mg-RE phases are almost unchanged, indicating that these phases have relatively good thermal stability. Generally speaking, the alloy containing fine, shear-resistant, evenly oriented (the planes parallel to the prismatic or basal planes of Mg matrix) and uniformly distributed particles, with adequate volume fraction, can exhibit a very high strength. The precipitation sequences in similar alloys, including Mg-Y-RE and WE43 during ageing, have been investigated in detail by other researchers [19,21–23], which involved Mg(SSSS)→β″→β′→β$_1$→β (SSSS: supersaturated solid solution). The β″ metastable phase is coherent with the α-Mg matrix in a $DO_{19}$ hexagonal structure (a = 0.642 nm, c = 0.521 nm). The metastable β′ precipitate, which is formed during ageing at 473–523 K, is semi-coherent, with the α-Mg in a base-centered orthorhombic structure. The β$_1$ phase is face-centered cubic structure (a = 0.74 nm). The sequence ends with the equilibrium β phase (fcc, a = 2.223 nm). In order to obtain high strength and high plasticity of the AZ31/WE43 bimetal composite, the optimized artificial ageing parameter, i.e., 483K for 48h, is obtained. In the case of the alloy aged at 483 K for 48 h, the fine, dense, and uniformly dispersed β′ and β$_1$ phases at the interface are the main precipitates, as shown in Figure 8. In previous studies [19,22], the β′ phase formed on the prismatic planes vertical to the basal plane provides the most effective obstacle to basal dislocation movement, which leads to the high strength of the T6-treated WE43 alloy. When the ageing time increases, from 24 h to 48 h, which reached the over-aged time at the transition interface, β′→β$_1$ transformation is generated, and β′ precipitates are larger and reduced. Thus, the shear strength of the interface decreased a little, due to the over-aging treatment. However, the more β′ precipitates generated after peak-ageing at 483 K for 48 h, leads to the high strength for BM WE43. However, the precipitates containing Al were unable to be found at interface layer after ageing treatment by the authors. It is possible that the amount of Al at interface is small, or the precipitates were too small to be found. After ageing, β′ and β$_1$ precipitates were generated in the interfacial zone and pinned dislocation slip and grain boundary movement. Thus, some kink belts and dimples were detected on the tripe junctions at grain boundary. Moreover, the strength of the interface was improved.

### 4.2. Fracture Behaviors

The shear strength of the interface zone, WE43, and AZ31, are summarized in Table 1. It is found that AZ31 can be strengthened by solution, but not by ageing treatment, while the interface zone and WE43 can be strengthened by solution and ageing treatment. Moreover, the WE43 is more significantly strengthened than the interface zone. In our previous study [11], the shear strength of the as-casting interface zone, WE43, and AZ31, was 108, 127, and 68 MPa, respectively. Cracks can initiate at the side of AZ31, due to it having the lowest shear strength. After solution treatment, the shear strength of the interface zone, WE43, and AZ31 was 118 MPa, 160 MPa, and 110 MPa, respectively, which are much higher than those of the as-cast samples. By solution treatment, the shear strength of the interface

zone, WE43, and AZ31 can be increased about 10, 33, and 42 MPa, respectively. The shear strength of the interface zone is similar as that of the basal material AZ31, which leads to the initial cracks of the interface generated in the zone of stress concentration, shown in Figure 10a,b.

**Table 1.** The shear strength of the interface zone, WE43, and AZ31 of the bimetal at as-cast, solution treatment, and aging treatment, for 24 h and 48 h.

| Treatment | Interface | AZ31 | WE43 | Reference |
|-----------|-----------|------|------|-----------|
| As-cast | 108 MPa | 68 MPa | 127 MPa | reference [11] |
| Solution treatment | 118 MPa | 110 MPa | 160 MPa | This study |
| Ageing treatment | 120 MPa | 101 MPa | 221 MPa | This study |

After ageing treatment, dispersed second phases are re-precipitated along the grain boundaries, and dense $\beta'$ phases are precipitated from $\alpha$-Mg matrix, which results in the increase of shear strength at the interface zone and that of WE43. The AZ31 cannot be strengthened by ageing treatment. The plasticity of the WE43 is significantly decreased, and that of the AZ31 is increased. Due to the lowest plasticity of WE43, the interfacial cracks of bimetal composites can be initialized near the WE43, and then propagated along the interface zone, which is in accordance with the observation of fracture morphologies shown in Figure 11. Thus, it seems that the interfacial cracks of bimetal composites are generated and propagated, depending on the cooperation of the strength and plasticity of the basal materials and interface.

## 5. Conclusions

In this study, the WE43/AZ31 bimetal composites have been prepared by inserting WE43 alloy rod into AZ31magnesium alloy melt, and subsequent T6 heat treatment. Interfacial microstructure evolution and bonding strength were evaluated and analyzed. Based on the observation and analyses, the conclusions can be summarized as follows:

(1) After solution treatment at 788 K for 12 h, most eutectic phases of matrix materials and interface are dissolved into $\alpha$-Mg matrix. Some high melting-point Mg-RE cuboid-shaped phases are distributed uniformly at the interface and WE43 matrix. The strength and plasticity are increased for the bimetal composites after solution treatment. Meanwhile, it is noted that the shear strength of AZ31 could improve about 40 MPa after solution treatment at 788 K for 12 h, due to the disappearance of the eutectic $Mg_{17}Al_{12}$ phase, and the reduction of casting defects.

(2) In order to achieve high strength and high plasticity of the bimetal composites, the optimized artificial ageing parameter of about 483 K for 48 h is obtained. After ageing treatment at 483 K for 48 h, $\beta'$ and $\beta_1$ precipitates are generated at the interface. The interfacial strength decreased a little for $\beta' \rightarrow \beta_1$ transformation, due to the over-aging treatment. However, the WE43 matrix achieves the highest strength, since more $\beta'$ phases can be generated at the peak-aged treatment. The AZ31 matrix achieves better plasticity with the increase of the ageing time. Therefore, the WE43/AZ31 bimetal composite will combine the high strength of WE43, and the high plasticity of AZ31 after heat treatment, exhibiting good application prospects for industry.

**Author Contributions:** D.X. and K.Z. conceived and designed the experiments; D.X. and K.Z. performed the experiments; D.X., K.Z. and H.L. analyzed the data; C.Y., H.L. and J.Z. contributed reagents/materials/analysis tools; D.X. wrote the paper.

**Funding:** This research was funded by The National Natural Science Foundation of China grant number 51671017, the Fundamental Research Funds for the Central Universities grant number FRF-GF-17-B3 and the fund of the State Key Laboratory of Solidification Processing in NWPU grant number SKLSP201835, and was fund by Beijing Laboratory of Metallic Materials and Processing for Modern Transportation.

**Conflicts of Interest:** The authors declare no conflict of interest.

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
