# Peer review of "Effect of Heat Treatment on Microstructure and Mechanical Properties of the AZ31/WE43 Bimetal Composites"

_metals, doi:10.3390/met8110971_

Round 1

Reviewer 1 Report

In this study, the mechanical properties of the AZ31/WE43 bimetal composites were reasonably explained from the viewpoint of the microscopic phase evolution caused by heat treatments. So, the manuscript is considered to be worth publishing in Metals. Before that, there are several minor revisions needed to address, as listed below.

1.       The experimental procedures part should be more improved (describe more in detail) to make it understandable for readers without referring to [11].

2.       P.4, line 131: Where do the lamellar-like (Al, Mg)2RE phases exist in the SEM or mapping photos? They are not visible, so should be clearly indicated by using arrows etc.

3.       The difference between Fig.9b-d is not clear. The caption of Fig.9 and its description in the text should be improved.

4.       Labels (a)-(d) must be placed on the photos in Fig.10.

5.        P.14, line 336: Delete one of “due to”.

Author Response

Thanks for your valuable suggestions. the manuscript has been revised and the answer to your question is in the attachment.

Reviewer 2 Report

The manuscript "Effect of heat treatment on microstructure and mechanical properties of the AZ31/WE43 bimetal composites" reports on the experimental study of the effect of heat treatment on the microstructure and mechanical properties of the 12 AZ31/WE43 bimetal composites.

From a general point of view, the topic of the manuscript is rather specialized. However, the results are certainly of interest for researchers working in the related field.

The introduction clearly states the aim of the work and sharply insert the work in the existing literature framework. The experimental section is complete and clear. The experiments are reliable and well-conducted. Worth of mention is the crossing of several experimental characterizations techniques for a complete understanding of the process. In this view, several experimental analysis are performed to draw a general scenario by connecting the experimental results to the process parameters. The discussions of the experimental results are, in general, reliable and well-founded, even if, in some points, a little bit too qualitative and general. The results are, however, original and of interest for researchers working in this field.

So, I think that the manuscript is suitable for publication on Metals after some minor revisions as noted in the following:

1)  During discussions, quantitative references and/or evaluations on the diffusion distance for Mg, RE, Al, Zn should be reported. Generally, in addition, the diffusion coefficient (or, also, the diffusion distance) presents an temperature activated behavior (D=D0exp(-Ea/kT)) characterized by an activation energy Ea (see, as examples: Acta Materialia 57, 4102 (2009); J. Appl. Phys. 101, 064306 (2007); Phys. Rev. Lett. 75, 2364 (1995); etc. are examples of the use of the thermally activated form of the diffusion or interdiffusion form of the diffusion coefficient in different fields). Using your temperature-dependent results, could be drawn some conclusion on Ea? It is very interesting from a quantitative point of view.  

2)  The authors often refer to the eutectic phase. A brief description of the phase diagram, highlighting the eutectic point, could help the reader.

3) Please, comment more sharply on the role of the interfacial defects on the mechanical properties after ageing. The authors often talk about grain boundaries, kinks, etc. However, a synthetic view is needed to draw the final conclusions.

4)  "Thus, it seems that the interfacial cracks of bimetal composites are generated and propagated depending on the cooperation of the strength and plasticity of the basal materials and interface." Please, discuss more extensively and quantitatively. I think that this an important conclusion which deserves more explanation.

Author Response

(The authors gave the same response as above.)

Reviewer 3 Report

This is a very interesting work that fits well within the scope of this Journal. However, to my opinion, some aspects need to be addressed prior to publication of this article. Minor revisions are due.

My comments/suggestions are given hereafter:

Line 84: The authors should give more information about the experimental procedure to make bimetal composite. For readers could be difficult obtained the reference 11 and then understand the problem.

Line 93: The authors should show the cut line of specimen.

Lines 184-190: The authors should format the text. The font size is not uniform.

Figure 11: The red text in Figures 11d and 11f is not clear the authors should improve improve these images.

Table 1: Format the text of table 1. Reduce the font size.

Author Response

(The authors gave the same response as above.)

Reviewer 4 Report

The paper shows interesting results, which can be potentially useful for technical applications. T. I will summarize below the revision I would suggest.

1.        Section 2.1: the authors describe the specimen with the reference 11. More detail and images of the specimens might be useful.

2.            Section 2.2: more detail about the interfacial shear test might be useful.

3.            Figure 10: the letters (a,b,c….) were lost

4             Error in reference row 184

Author Response

(The authors gave the same response as above.)
